# Tree species traits affect which natural enemies drive the Janzen-Connell effect in a temperate forest

Shihong Jia [1,2,3], Xugao Wang[1]*, Zuoqiang Yuan [1], Fei Lin[1], Ji Ye[1], Guigang Lin[1], Zhanqing Hao[1,4] & Robert Bagchi [3]

A prominent tree species coexistence mechanism suggests host-specific natural enemies inhibit seedling recruitment at high conspecific density (negative conspecific density dependence). Natural-enemy-mediated conspecific density dependence affects numerous tree populations, but its strength varies substantially among species. Understanding how conspecific density dependence varies with species' traits and influences the dynamics of whole communities remains a challenge. Using a three-year manipulative community-scale experiment in a temperate forest, we show that plant-associated fungi, and to a lesser extent insect herbivores, reduce seedling recruitment and survival at high adult conspecific density. Plant-associated fungi are primarily responsible for reducing seedling recruitment near conspecific adults in ectomycorrhizal and shade-tolerant species. Insects, in contrast, primarily inhibit seedling recruitment of shade-intolerant species near conspecific adults. Our results suggest that natural enemies drive conspecific density dependence in this temperate forest and that which natural enemies are responsible depends on the mycorrhizal association and shade tolerance of tree species.

[1] CAS Key Laboratory of Forest Ecology and Management, Institute of Applied Ecology, Chinese Academy of Sciences, Shenyang 110016, China. [2] University of Chinese Academy of Sciences, Beijing 100049, China. [3] Department of Ecology and Evolutionary Biology, University of Connecticut, 75N. Eagleville Road, Storrs, CT 06269, USA. [4] Research Center for Ecological and Environmental Sciences, Northwestern Polytechnical University, Xi'an 710072, China. *email: wangxg@iae.ac.cn

Natural enemies (e.g. fungal pathogens, insects, and mammalian herbivores) have long been postulated to play crucial roles in determining the diversity of plant communities[1–4]. The Janzen-Connell hypothesis[3,4] provides one mechanism, suggesting that host-specific natural enemies are concentrated where their host plant species occur at high densities (e.g. close to conspecific maternal adults), suppressing recruitment and survival of their host plants (negative conspecific density dependence)[3–6]. Such self-inhibition by conspecifics puts rare species at an advantage and facilitates plant species coexistence and diversity. Several empirical studies have reported that seedling recruitment and survival decrease with the density of conspecific neighbors, but are relatively unaffected by heterospecifics[7–9]. There is also increasing experimental evidence that insect herbivores and, pathogenic fungi in particular, are responsible for mediating the conspecific density dependence observed in several plant populations[5,10–15]. Despite growing consensus that natural enemies regulate plant population dynamics, however, the extent to which this process contributes to diversity at the community scale remains poorly known (but see refs. [16,17]).

The strength of conspecific density dependence varies considerably among species[16,18,19], which is one factor complicating assessments of its contribution to plant community diversity. This variation may be correlated with plant species' traits[20–22], which potentially provides a basis for extrapolation to the community-scale. There is some evidence that slow-growing, shade-tolerant tree species are less affected by conspecific neighbors than shade-intolerant species[19–21,23], potentially because shade-tolerance is associated with greater investment in defense or storage, and hence lower susceptibility to natural enemies[24,25]. Additionally, types of mycorrhizal association (e.g. arbuscular mycorrhizal, AM or ectomycorrhizal, EM) differ in their effectiveness in protecting plant roots from antagonists[26,27], which could translate into variation in susceptibility of hosts to natural enemies and hence density dependence. For example, a continental-scale study reported that EM seedlings respond positively to the density of conspecific trees, while AM seedlings respond negatively[28]. However, the relative importance of different traits in modifying the strength of natural-enemy-mediated density dependence in intact communities remains largely untested.

The strength of conspecific density dependence may also vary among ecosystems. The Janzen-Connell hypothesis originally postulated that natural enemy-induced conspecific density dependence should be more negative in tropical than in temperate systems[1,3,4]. Nevertheless, the relationship between latitude and the strength of conspecific density dependence remains disputed[29–33]. Inferences based on non-manipulative surveys are sensitive to analytical methods, with both marked weakening of conspecific density dependence with latitude[31,32] and no relationship reported using the same datasets[33]. A meta-analysis of manipulative experiments of individual plant populations reported no consistent relationship between conspecific density dependence and latitude[30]. The strength of conspecific density dependence varies greatly among experiments and species so community-wide experiments have proved useful tools in determining the importance of density dependence for diversity in tropical forests[16,17]. However, community-scale experimental manipulations of natural enemies, equivalent to those performed in tropical forests (e.g. refs. [16,17]), have not been attempted in less species-rich, temperate forests.

We report on a field experiment in an old-growth temperate forest in northeast China to test for conspecific density dependence at the community scale, to identify the groups of natural enemies that may be responsible and to examine whether variation in the strength of density dependence among plant species is associated with two important plant traits: shade tolerance and type of mycorrhizal association. We test two hypotheses: (1) Plant-associated fungi and insect herbivores are critical drivers of conspecific density dependence and hence seedling diversity and composition in this temperate forest, and (2) Species varying in mycorrhizal association and shade-tolerance differ in their sensitivity to enemy-mediated density dependence. Specifically, based on the literature we predict that AM trees are more sensitive to fungal-mediated conspecific density-dependence than EM trees[28], and that conspecific density dependence mediated by both plant-associated fungi and insects are greater for shade-intolerant than shade-tolerant species[19,20,23]. We manipulated the access of three groups of natural enemies, large herbivores (using fences), insect herbivores (using insecticide) and plant-associated fungi (using fungicide), to seedlings in 180 1 m$^2$ quadrats (Supplementary Fig. 1). We followed seedling recruitment and survival in these quadrats from 2015 to 2017. We focus on the earliest life stages of trees because they are most strongly affected by natural enemies[16,34,35]. We quantify conspecific and heterospecific adult densities by summing the inverse-distance weighted basal area of all trees >5 cm diameter at breast height (DBH) within a 20 m radius of each quadrat. We use the 20 m radius because intraspecific interactions are often undetectable beyond this distance[36,37]. Our results indicate that plant-associated fungi have a negative effect on seedling recruitment and survival at high conspecific adult density. Insect herbivores also inhibit seedling recruitment near conspecific adults, but they do not affect seedling survival. Additionally, which groups of natural enemies impact seedling recruitment near conspecific adults depends on two important tree species traits (i.e. mycorrhizal association and shade-tolerance). Plant-associated fungi have a stronger negative effect on seedling recruitment for EM-associated and shade-tolerant species than AM-associated and shade-intolerant species. In contrast, seedling recruitment is more strongly inhibited by insect herbivores near conspecific adults for shade-intolerant species than shade-tolerant species.

## Results

**Community-wide conspecific density dependence.** Across the 180 quadrats, 3929 tree seedlings recruited from 16 species (Supplementary Table 1) in the censuses from 2015 to 2017. The overall mortality rate through the growing seasons was 29.4%.

Seedling survival decreased with conspecific adult densities, although not significantly (binomial generalized linear mixed-effects model [GLMM]: $z = -1.53$, $P = 0.13$). Protection from fungi increased both recruitment and survival of seedlings when conspecific adult densities were high (Fungicide × conspecific adult density interactions, recruitment: Poisson GLMM, $z = 4.87$, $P < 0.001$; survival: binomial GLMM, $z = 3.18$, $P = 0.001$; Fig. 1). Suppression of insects also increased seedling recruitment at high conspecific adult densities (Poisson GLMM: Insecticide × conspecific adult density interaction, $z = 2.19$, $P = 0.029$), but did not affect seedling survival appreciably. The effect of large vertebrates was not estimated to be large in any of the models (Recruitment: Poisson GLMM, $z = -0.23$, $P = 818$; Survival: binomial GLMM, $z = -0.22$, $P = 0.823$; Fig. 1), so was not considered further in the analyses.

**Mycorrhizal association and conspecific density dependence.** A total of 1926 individuals (49% of all seedlings) recorded were from five species of EM trees, most of which (93% of EM seedlings) were shade tolerant. The remaining seedlings belonged to 11 AM species, which included both shade-tolerant (seven species, 757 individuals) and shade-intolerant (four species, 1246 individuals) species. The influence of natural enemies on seedling

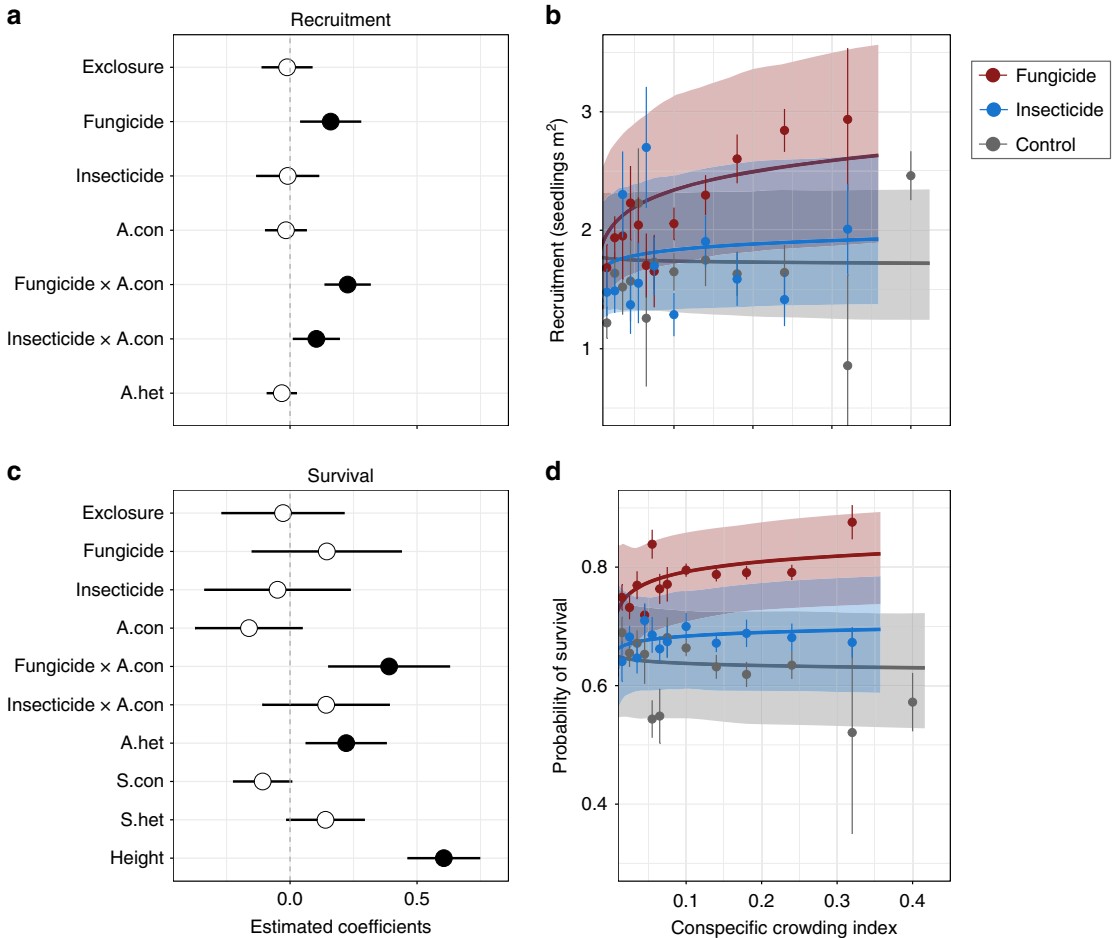

**Fig. 1 Effects of natural enemies and neighborhoods on seedling recruitment and survival. a, c** Parameter estimates from Generalized Linear Mixed Effects Models (GLMM) fitted to seedling recruitment ($n = 1019$) and seedling survival ($n = 3929$) data from all species in the seedling assemblage. The models estimated the effects of pesticide application, conspecific adult density (A.con) and the interactions between them. Heterospecific adult density (A. het) and exclosure treatment were included as covariates in both models. The survival model also included terms for densities of conspecific and heterospecific seedlings (S.con and S.het) and the effect of seedling height (Height). Solid points indicate parameter estimates that are significantly different ($P < 0.05$) from zero (dashed lines) and bars indicate 95% confidence intervals. **b, d** The relationships between neighboring conspecific adult densities and seedling recruitment and seedling survival under each pesticide treatment. Lines and shaded polygons indicate the GLMM predictions and their 95% confidence intervals. Dots and bars represent the mean and SE of the observed values, which were calculated by adding model residuals to the predicted values. We averaged the observed values within bins to facilitate visualization (with bin width increasing as the conspecific crowding index decreases). Source data are provided as a Source Data file.

recruitment and survival near adult conspecifics differed between EM and AM tree species. The relationship between conspecific adult density and recruitment of EM species was significantly more positive in the fungicide treatment than in either the control or insecticide treatments (Poisson GLMM: Fungicide × conspecific adult density interaction, $z = 2.89$, $P = 0.004$; Fig. 2). Survival of EM species decreased as adult conspecific density increased (binomial GLMM: $z = −2.63$, $P = 0.008$), but fungicide did not appreciably ameliorate that effect (binomial GLMM: $z = 1.23$, $P = 0.220$; Fig. 3). In contrast to EM species, the relationship between conspecific adult density and recruitment of AM species was significantly less affected by fungicide treatment (Poisson GLMM: significant AM × Fungicide × conspecific adult density interaction, $z = −2.22$, $P = 0.027$). Additionally, survival of AM species was significantly less dependent on conspecific adult density (binomial GLMM: AM × conspecific adult density interaction, $z = 2.54$, $P = 0.011$). The effects of insecticide treatment on recruitment and survival did not differ between mycorrhizal association types (Fig. 3).

**Shade tolerance and conspecific density dependence.** Ten shade-tolerant species (2540 individuals) accounted for 65% of all seedlings. Recruitment of shade tolerant species increased with adult conspecific density, and this positive relationship strengthened significantly when fungicide was applied (Poisson GLMM: $z = 2.52$, $P = 0.012$; Fig. 2). In contrast, recruitment of shade intolerant species was slightly negatively associated with adult conspecific density and was unaffected by fungicide addition (Fig. 2). These differences between shade tolerant and intolerant species in terms of the effects of conspecific adult density and fungicide addition on recruitment were statistically significant (Poisson GLMMs: Shade-intolerance × conspecific adult density interaction, $z = −4.31$, $P < 0.001$; Shade-intolerance × Fungicide interaction, $z = −2.84$, $P = 0.005$). Nevertheless, the interactive effect of adult conspecific density and fungicide on seedling recruitment was similar for shade tolerant and shade intolerant species (i.e. the three-way interaction was non-significant). Insecticide, on the other hand, increased recruitment of shade-intolerant seedlings near conspecific adults significantly

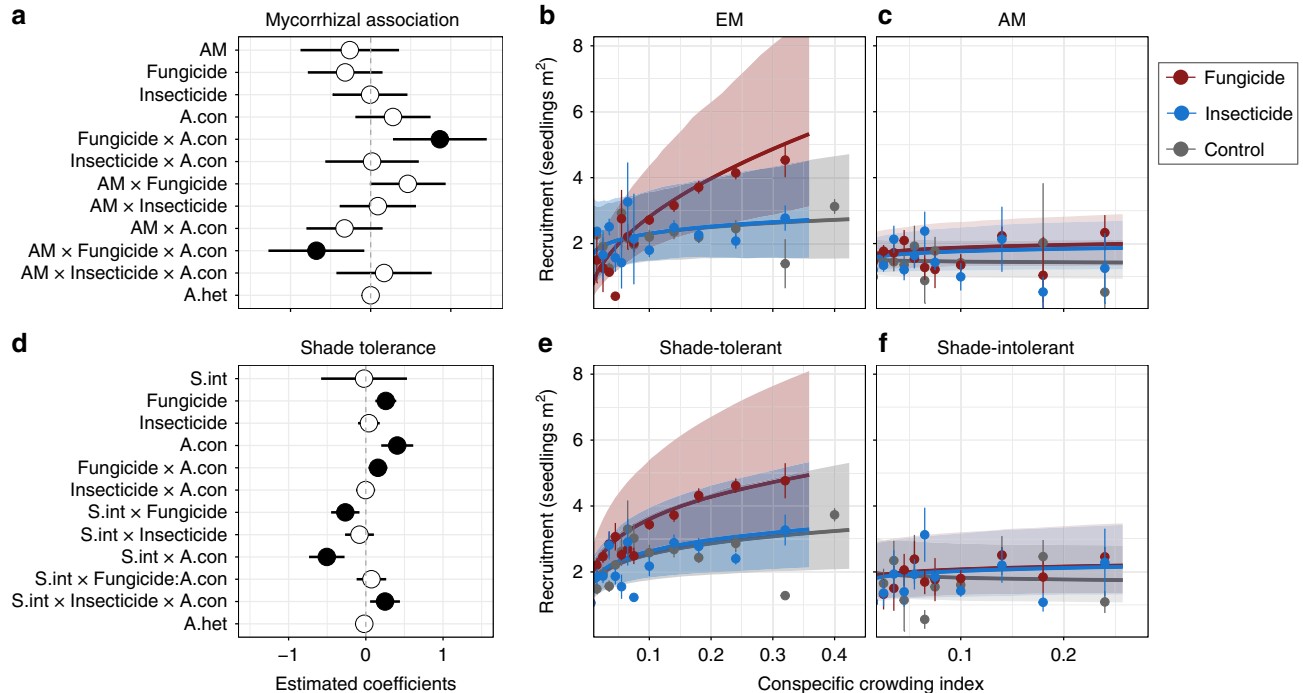

**Fig. 2 Effects of natural enemies, conspecific adult density, and plant species' mycorrhizal association type or shade tolerance status on seedling recruitment. a, d** Parameter estimates from Generalized Linear Mixed Effects Models (GLMM) fitted to estimate the effects of pesticide application, conspecific adult density (A.con), mycorrhizal association, or shade tolerance (shade-intolerance coefficient indicated as S.int) and the two- and three-way interactions. Solid points indicate parameter estimates that are significantly different ($P < 0.05$) from zero (dashed lines) and bars indicate 95% confidence intervals. **b, c** The relationships between conspecific neighboring adult densities and seedling recruitment under each treatment for EM and AM species, respectively. **e, f** The relationships between conspecific neighboring adult densities and seedling recruitment under each treatment for shade-tolerant and shade-intolerant species. Lines and shaded polygons indicate the GLMM predictions and their 95% confidence intervals. Dots and bars represent the mean and SE of the observed values, which were calculated by adding model residuals to the predicted values. We averaged the observed values within bins to facilitate visualization (with bin width increasing as the conspecific crowding index decreases). Source data are provided as a Source Data file.

more than recruitment of shade tolerant species (Poisson GLMM: Shade-intolerance × Insecticide × conspecific adult density interaction, $z = 2.52$, $P = 0.012$). Shade tolerance did not influence the response of seedling survival to either conspecific adult density, biocides or their interactions (Fig. 3).

**Seedling diversity and composition.** Despite the effects of pesticide application on recruitment and survival, seedling diversity (Shannon's diversity index) was unaffected by suppressing natural enemies (linear mixed-effects models of Shannon diversity index: Fungicide: $t_{336} = 0.98$, $P = 0.330$; Insecticide: $t_{336} = 0.16$, $P = 0.875$). Spraying fungicide, however, did influence species composition, significantly decreasing the dissimilarity between seedlings and surrounding adult trees (linear mixed-effects model: $t_{336} = -3.78$, $P < 0.001$; Fig. 4). Thus, it appears that the fungi lead to recruitment of species that are dissimilar to nearby adult trees, even if diversity at the plot scale is not affected. Insects do not appear to affect the species composition of recruits in this forest.

## Discussion

The deleterious effects of natural enemies on seedling recruitment and survival, assessed at the community scale, increase with the local density of conspecific adults. While this pattern is consistent with the Janzen-Connell hypothesis, the relationship between adult density and seedling survival is weak in plots unprotected from natural enemies. Suppressing natural enemies does not reduce tree species diversity, which is consistent with the lack of

dependence of seedling survival on local conspecific seedling density, regardless of biocide treatment. Instead, fungicide application lead the species composition of seedlings to more closely resemble that of the surrounding adults, which could result from fungi increasing mortality of seedlings near conspecific adults. Thus, our results suggest that plant-associated fungi suppress the recruitment of tree species dominating the nearby canopy, which will increase turnover of tree species and could increase diversity.

While the evidence that natural enemies reduce seedling performance at high conspecific density in our system is in line with previous pesticide application experiments in field conditions in tropical forests[15–17], the strength of conspecific density dependence observed in this temperate forest appears much weaker. Additionally, in contrast to previous community-wide manipulations of natural enemy communities (e.g. refs. [16,17]), we found no evidence that attack by either plant-associated fungi or insect herbivores is related to the density of conspecific seeds and seedlings, or that it increase seedling diversity. Instead, our results suggest that density of adult conspecifics is a better predictor of natural enemy impacts on recruitment and survival of seedlings. These natural enemy impacts appear to be important, even though the net effect of conspecific density is only weakly negative in plots when natural enemies are unmanipulated. Both recruitment and survival go from negatively to positively related to conspecific adult density when fungi and, to a lesser extent, insects are excluded. As a result, seedling communities are less similar to adults than when fungi are suppressed. Several field and shade-house experiments involving small numbers of tree species

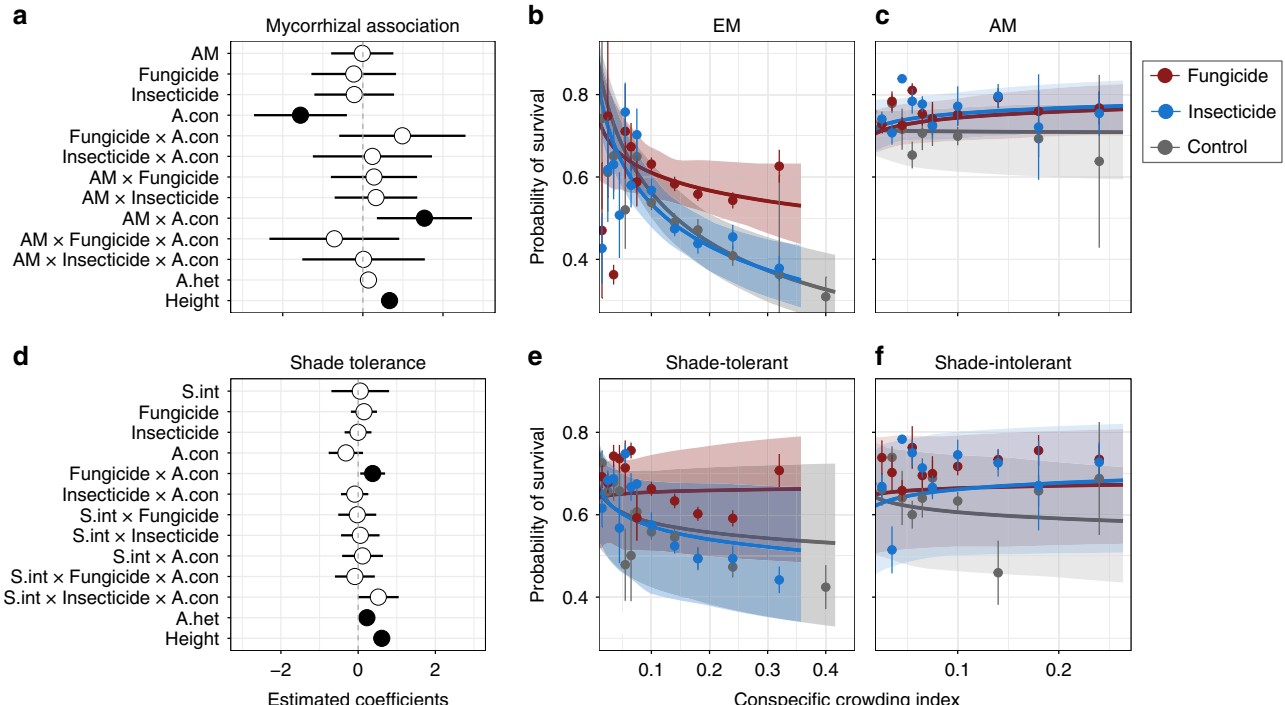

**Fig. 3 Effects of natural enemies, conspecific adult density, and plant species' mycorrhizal association type or shade tolerance status on seedling survival. a, d** Parameter estimates from Generalized Linear Mixed Effects Models (GLMM) fitted to estimate the effects on survival of pesticide application, conspecific adult density (A.con), mycorrhizal association, or shade tolerance (shade-intolerance coefficient indicated as S.int) and the two- and three-way interactions. Solid points indicate parameter estimates that are significantly different ($P < 0.05$) from zero (dashed lines) and bars indicate 95% confidence intervals. **b, c** The relationships between neighboring conspecific adult densities and seedling survival under each treatment for EM and AM respectively. **e, f** The relationships between neighboring conspecific adult densities and seedling survival under each treatment for shade-tolerant and shade-intolerant species. Lines and shaded polygons indicate the GLMM predictions and their 95% confidence intervals. Dots and bars represent the mean and SE of the observed values, which were calculated by adding model residuals to the predicted values. We averaged the observed values within bins to facilitate visualization (with bin width increasing as the conspecific crowding index decreases). Source data are provided as a Source Data file.

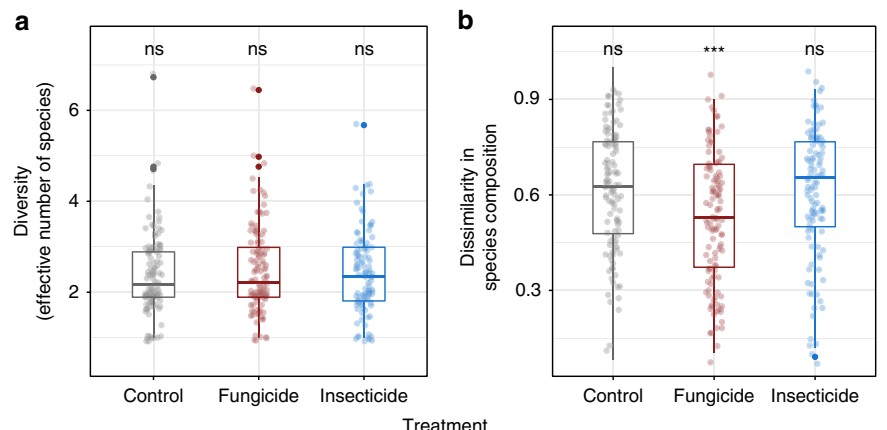

**Fig. 4 Effects of natural enemies on seedling diversity and composition.** Suppression of plant-associated fungi and insect herbivores on **a** Shannon diversity of seedlings (represented by the effective number of species), and **b** the abundance-weighted Bray-Curtis dissimilarity between the species composition of seedlings and surrounding adults. We pooled the data from two censuses for each treatment ($n = 120$). Each point shows an observed value for seedling diversity or dissimilarity in each quadrat. The boxes show the first and third quartiles, the lines within the boxes indicate the medium, the whiskers outside the boxes are the smallest and largest values within 1.5× interquartile range. Asterisks at the top of boxes indicate significant pairwise difference ($P = 0.05$ level) among treatments, which were estimated with linear mixed-effects models. Source data are provided as a Source Data file.

have indicated that performance is inhibited near conspecific adults as a result of plant-associated fungi[5,11,15,38,39] and insect herbivores[15,40]. Our results extend the experimental evidence that seedling performance is suppressed by adult conspecifics to a community scale.

While the presence of conspecific density dependence is well-established in plant communities, recent studies have observed that the strength of density dependence varies widely across species with contrasting traits[20,21]. Our analyses identified that recruitment of EM-associated and shade-tolerant species is more

strongly inhibited by adult conspecifics, although the correlation between mycorrhizal association and shade-tolerance (most EM seedlings are shade tolerant), hinders any definite distinction between the effects of the two traits. Suppression of plant-associated fungi increase seedling recruitment of EM species in areas of high conspecific adult density. While there is little difference between EM and AM species in the effects of adults on seedling recruitment, fungi do not appear to be responsible for conspecific density dependence in AM species. Interestingly, previous studies have suggested that EM seedlings should preferentially recruit near conspecific adults[28,41,42], in contrast to our results. One possible explanation for this disagreement is that we concentrated on the earliest stages in the trees' lives, before mycorrhizal associations are generally established. At this stage, the mycorrhizal colonization rate is much lower (Supplementary Fig. 2) compared to that of mature trees[43]. Adult conspecifics provide a source of EM fungi that facilitate nutrient uptake and may protect seedlings from antagonistic fungi[27,28]. The benefits of adult proximity for older saplings able to establish mycorrhizal associations may outweigh the costs from heightened exposure to fungal pathogens, especially under lowlight, understorey conditions. Pre-mycorrhizal seedlings will only incur the costs of proximity to adult conspecifics, however, perhaps generating the relationships we observed.

Species' shade-tolerance is also related to the strength of conspecific density dependence in our analyses, as well as the identity of the natural enemies associated with density dependence. In control plots, the increase in recruitment in areas of high conspecific adult density probably just reflects greater seed input, given that dispersal limitation is strong in our forest[44] and leads to an overall positive relationship between conspecific adult density and seed numbers (Supplementary Fig. 3). Importantly, seed input is much higher near conspecific adults for shade-tolerant than shade-intolerant species (Poisson GLMM: Shade-intolerance × conspecific adult density, $z = -2.25$, $P = 0.024$. Supplementary Fig. 3). Fungicide addition strengthens the relationship, suggesting that fungi reduce seedling recruitment of shade-tolerant species near adult conspecifics. In contrast, seedling recruitment of shade intolerant species is not concentrated near adult conspecifics, perhaps reflecting greater dispersal ability and a need for these species to disperse into high light areas as well as the fact that the canopy is mostly dominated by shade-tolerant species (Supplementary Fig. 4). Interestingly, insecticide addition increases recruitment of shade intolerant species near conspecific adults, whereas fungicides have no effect. These results suggest that shade tolerance predicts the type of natural enemies that plant species are susceptible to. Shade-tolerant species tend to be most often attacked by aggressive, necrotrophic fungal pathogens, whereas shade intolerant species are often infected by less aggressive biotrophic pathogens[45]. Necrotrophs may kill large numbers of susceptible plants and their distributions can be highly spatially structured[46], making them prime candidates for mediating plant-soil feedbacks close to adult conspecifics. Shade-intolerant species generally have lower wood density and smaller seed size[21,47,48], traits associated with lower investment in defense against insect attack than shade-tolerant species[24,25]. Previous studies have highlighted that shade-tolerance is associated with the extent that species experience conspecific density dependence[19–21,49], but we believe that this is the first time that shade-tolerance has been linked to the class of natural enemies mediating conspecific density dependence. We caution that these relationships are phenomenological and because suites of traits tend to be inter-correlated, life-history traits other than shade-tolerance might be truly driving them. However, because different natural enemies vary in their capacity to exert density dependent control of plant populations, for

example due to variation in their mobility, host-specificity and virulence, the identities of natural enemies may explain variation of density dependence among tree species[50,51]. Investigating how the drivers of conspecific density dependence vary with life-history traits could provide a fruitful avenue for future research.

Although much of the evidence that natural enemies cause conspecific density dependence in tree seedlings has come from tropical forests[5,15,16,30], there are a growing number of studies in temperate systems that also provide evidence for such effects[11,23,52,53]. While the strength of conspecific density dependence observed in this temperate forest is weaker than that observed in tropical studies (e.g. refs. [8,16,17,19]), our results suggest that the causes of density dependence may be consistent across latitudes. Several studies in tropical forests have suggested that while insects influence seedling recruitment and might mediate density dependence in some species[15,40], plant-associated fungi play a bigger role in mediating conspecific density dependence at the community scale[16,17]. Similarly, although insects contribute to conspecific density dependence in our experiment, suppressing fungi has a greater effect on seedling recruitment and survival than insecticide application at the community scale. While fences are effective at excluding large herbivores such as wild boars and roe deer (Supplementary Fig. 5) from our plots, our results do not support a role of large mammalian herbivores in density dependent control of plant populations, in agreement with previous studies[54,55]. The low host-specificity of many large herbivores[56] may explain why they do not appear to contribute to density dependence in our experiment. Alternatively, the low number of replicate exclosure plots (three pairs of fenced and unfenced plots) may limit our ability to detect impacts of large herbivores on plant communities. Further, better replicated experiments that exclude large herbivores may be necessary to determine their role in shaping plant communities in these ecosystems. Although further work is necessary to determine the identity and host-specificity of insect herbivores and plant-associated fungi, perhaps using molecular techniques (e.g. refs. [57,58]), on the basis of the evidence presented here, insects and plant-associated fungi appear to play a greater role in mediating conspecific density dependence than large herbivores.

Overall, our field experiment provides evidence of a community-wide Janzen-Connell effect in this temperate forest. Comparison of these results to those from similar experiments in tropical forests[16,17] suggests that causes of conspecific density dependence (e.g. plant-associated fungi and insect herbivores) may be consistent across latitudes. Additionally, our results indicate that the strength of density dependence largely depends on the association between the types of natural enemies and the mycorrhizal association type or shade tolerance of tree species. Although natural-enemy-mediated density dependence does not translate into increased seedling diversity, our results suggest that plant-associated fungi play a critical role in shaping species composition and turnover in our temperate forest. Notably, our study highlights the need to consider multiple types of natural enemies and traits of plant species to improve our understanding of the mechanisms of conspecific density dependence and species coexistence.

## Methods

**Study site**. This study was conducted within the Changbai Mountain National Nature Reserve in northeast China (42°23' N, 128°05' E, ~ 800 m a.s.l.). The climate is characterized by an annual mean temperature of 2.8 °C (−13.7–19.6 °C) and average annual precipitation of ~700 mm[59], mostly as rain from June to September and snow otherwise, with the ground covered by snow from November to April. The vegetation is dominated by broad-leaf and mixed forest over slightly acidic (mean pH = 5.45) dark brown forest soil. Dominant tree species include *Tilia amurensis*, *Fraxinus mandschurica*, *Quercus mongolica*, and *Pinus koraiensis*.

**Experimental design.** We established three 55 m × 50 m blocks, separated by at least 200 m, within an old growth temperate forest in September 2014 (experimental layout in Supplementary Fig. 1). The central area (35 m × 30 m) of each block was split into two equally sized plots, with one plot randomly allocated to a large herbivore exclosure treatment and enclosed with a 1-m high fence constructed of 4–5 steel wires vertically spaced by 0.2 m. Within each plot, we demarcated 50 1 × 1 $m^2$ seedling quadrats on a grid, with quadrats separated by ~2 m and avoiding large roots. We randomly selected thirty of these quadrats for use in the experiment described here. These 30 quadrats were divided equally and randomly among three pesticide treatments: fungicide, insecticide, and control.

To confirm that our fences were effective at excluding large herbivores, we installed infrared cameras (Bestguarder SG-990V, Shenzhen, China) at three locations within and outside each fence between September 2015 and September 2017. In addition, in each census we examined all quadrats for evidence of large mammals (e.g. footprints or feces).

We initiated pesticide treatments in July 2015 and continued until October 2017. Plants were protected from true fungi and oomycetes through a combination of two fungicides, Amistar (Syngenta Ltd, active ingredient Azoxystrobin) and Ridomil Gold MZ 68 (Syngenta Ltd, active ingredients metalaxyl and mancozeb). Both fungicides have low toxicity to non-target organisms and have been reported to have little effect on mycorrhizal colonization[60,61]. Insects were deterred by applying a broad spectrum, systemic insecticide, Engeo (Syngenta Ltd, active ingredient Thiamethoxam). We applied pesticides biweekly except during the winter. Pesticide concentrations followed the manufacturer's guidelines (Fungicide: 0.005 g of Amistar + 0.25 g of Ridomil Gold; Insecticide: 0.0025 ml of Engeo, each dissolved in 50 ml of water). Control quadrats were sprayed with an equal volume of water.

We conducted censuses of woody plants in June 2015, 2016, and 2017. All woody plants <1 cm DBH (Diameter at Breast Height) were tagged, mapped and identified to species in the 180 quadrats. In September of each year, we checked the status (survival/dead) of each existing seedling. We focused on survival status in the first few months after recruitment (seed-to-seedling transition), because seedlings are most vulnerable to natural enemies during this stage[16,62]. We calculated recruitment as the number of new seedlings >1 cm tall in each quadrat in every census. Including tiny seedlings would allow us to indirectly measure the effects of natural enemies on seedling survival during the seed to seedling transition. We did not evaluate the effect of seed abundance on recruitment because the majority of tree species recruit from seed bank rather than seed rain, and the time germinating from seeds varies from months to years across species. In June 2017, we identified all trees (DBH > 5 cm) within each block and recorded their species identity, DBH and distance to each quadrat within 20 m.

**Statistical analysis.** Analyses of seedling recruitment and survival only considered tree seedlings recruited after pesticide applications were initiated. We used the crowding index (i.e. A.con and A.het) to represent the conspecific and heterospecific adult densities, which were calculated by summing the inverse-distance weighted basal areas of all conspecific and heterospecific adults within a 20-m radius of each quadrat[8,63]. The range of the conspecific crowding index within the 20 m radius of each quadrat was from 0 to 0.42. We also calculated the conspecific (S.con) and heterospecific seedling densities (S.het) as the number of seedlings within each quadrat (S.con: 0–27 seedlings $m^{-2}$; S.het: 0–44 seedlings $m^{-2}$).

We used generalized linear mixed-effects models (GLMM) to evaluate the effects of enemy types and neighboring plants on seedling recruitment and survival. All continuous explanatory variables were log-transformed and then standardized (i.e. by subtracting the mean and dividing by the standard deviation) to facilitate the direct comparison of the relative importance of each variable[64]. We included pesticide treatment, exclosure treatment and conspecific adult density (i.e. A.con) and the two- and three-way interactions among them as fixed factors in the full model for recruitment. Heterospecific adult density (A.het) was included as a covariate in the recruitment model. For survival, we added covariates describing seedling neighborhoods (conspecific seedling density-S.con and heterospecific seedling density-S.het) in addition to all the covariates included in the recruitment model. We did not include seedling densities in recruitment analyses because the number of recruits were confounded with the seedling densities. Seedling height was also included as a covariate to account for the potential positive relationship between survival and seedling size[65]. To evaluate the contribution of two- or three-way interactions in explaining seedling recruitment and survival, the full models considering all interactions were compared with all possible reduced models from which one or more interactions were removed using likelihood ratio tests ($\chi^2$ with degree of freedom of the differences in the number of parameters)[66]. The models assumed Poisson and binomial error distributions for recruitment and survival respectively, and we confirmed that model residuals were not overdispersed. To account for spatial and temporal autocorrelation between closely located plots and censuses in the data, we included block and census as categorical fixed effects[14]. We also considered quadrat and species identity as random intercepts and allowed the effect of census to vary among quadrats as a random effect. We did not include random slopes for different species because preliminary analyses indicated that allowing slopes to vary among species led to less parsimonious models (higher AIC) than intercept-only models, and the coefficients did not change qualitatively.

To explore correlates of interspecific variation in covariate effects, we categorized each tree species by shade tolerance (shade-tolerant or shade-intolerant) and mycorrhizal association (arbuscular mycorrhizal-AM or ectomycorrhizal-EM; Supplementary Table 1). We considered three-way interactions between mycorrhizal association or shade tolerance, pesticide treatment, and conspecific adult density as fixed effects. We did not consider S.con and S.het in these models as preliminary analyses suggested their effects on seedling survival were negligible.

Two species (T. amurensis and F. mandschurica) accounted for 74% of all seedlings. To evaluate the influence of these two species on the community-wide and correlates of interspecific variation results, we conducted a sensitivity analysis. We randomly removed 25, 50 and 75% of individuals of T. amurensis and F. mandschurica from the community data sets (while including all individuals of the other species) and refitted the models, repeating this procedure 999 times. We compared parameter estimates from these simulations to the models including all seedlings. Despite their numerical dominance of the seedling assemblage, our results were insensitive to removing random fractions of individuals of these two species (Supplementary Figs. 6–11).

Finally, we examined whether suppression of plant-associated fungi and insect herbivores altered species diversity and composition in the seedling assemblage. We estimated the diversity of seedlings in each plot and census using Shannon's diversity index (H)[67]. Turnover in species composition between conspecific adult neighbors within 20 m of each quadrat and the seedlings within the quadrat was quantified using the Bray-Curtis index[68]. This metric quantifies the effects of biocide treatments on the compositional development of the seedling community along the trajectory towards the surrounding adults. We used linear mixed-effects models with pesticide treatment as a fixed effect. Census was also included as a covariate to account for the temporal autocorrelation. We included random intercepts for plots.

All the data analyses were conducted using R v 3.4.2[69] using the package "lme4"[70].

**Reporting summary.** Further information on research design is available in the Nature Research Reporting Summary linked to this article.

## Data availability
The datasets analyzed in this study are archived on Figshare (https://doi.org/10.6084/m9.figshare.11300534). The source data underlying Figs. 1–4, Supplementary Figs. 2–4 and Supplementary Figs. 6–11 are provided as a Source Data file.

## Code availability
The code supporting the results are archived on Figshare (https://doi.org/10.6084/m9.figshare.11300534).

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

## Acknowledgements

We thank S.J. Wright, M.S. Luskin, D.J. Johnson, S.J. Davies, and V. Milici for valuable comments. We also thank B. Wu, J. Hu, Z. Mao, S. Fang, and members of Wang lab for field assistance. Portions of this work benefited from the 2016, 2017, and 2018 ForestGEO workshops attended by S.Jia. This work was financially supported by the Strategic Priority Research Program of the Chinese Academy of Sciences (Grant XDB31030000), the National Natural Science Foundation of China (Grant 31722010), the Key Research Program of Frontier Sciences, Chinese Academy of Sciences (Grant ZDBS-LY-DQC019), the LiaoNing Revitalization Talents Program (Grant XLYC1807039), and the K. C. Wong Education Foundation.

## Author contributions

X.W. and S.J. conceived the idea and designed the research. S.J., X.W., Z.Y., F.L., J.Y., G.L., and Z.H. collected the data. S.J. and R.B. conducted the data analyses and wrote the first draft. All authors contributed to the final manuscript.

## Competing interests

The authors declare no competing interests.
