## [Peer Review File · Nature Communications]

Reviewers' comments:

Reviewer #2 (Remarks to the Author):

I have reviewed this paper for another Nature journal previously and I am happy to find that the authors have corrected most of the issues I raised previously. The authors also tried their best to address the comments from other referees.

This study tests the strength of conspecific negative density dependence (CNDD) in a temperate forest, identify which groups of natural enemies are more critical drivers of CNDD, and whether mycorrhizal association and shade-tolerance could predict enemy-mediated CNDD. They show in field manipulated experiments that fungicide, and less importantly insecticide, promoted seedling recruitment and survival, and these effects were significantly associated with conspecific density. The authors also show that EM species and shade-tolerant species were more sensitive to adult conspecifics. All these findings are consistent with density dependent accumulation of natural enemies.

While similar experiments have been conducted in tropical forests, which may there are many things I like about this manuscript. This study provides a good investigation on CNDD in a temperate forest, with detailed dataset from a 3-year manipulate experiment, which could be a nice complement to the existing literature that mainly focused on tropics and subtropics. By manipulating the access of different types of natural enemies, the authors actually open the “black box” of CNDD to reveal the relative importance of various drivers.

There are still a few things that I like less. First, as Reviewer 1 pointed out, although the natural enemy effects on seedling performance are significantly correlated with host density (i.e. density dependence), the overall effects of host density in control treatments (i.e. under natural conditions) are neutral for both survival and recruitment. That is to say, it's not *negative* density dependence. This does not change the main conclusions of the study, but I would suggest the authors to reevaluate the usage of the term “conspecific negative density dependence”. Maybe just “conspecific density dependence” would be more appropriate in this case.

Second, the authors found that ectomycorrhizal species are more sensible to negative plant-soil feedbacks near conspecific adults, which is conflicting with many other recent studies that found EM species were less negatively affected by pathogens related to AM species (e.g. Bennett et al. Science 355: 181-184, 2017). Although the authors discussed this and provided on possible explanation, this still sounds unconvincing to me. Mycorrhizal associations could be established on seedling roots very quickly, especially with high conspecific adult density.

Third, the two hypotheses tested in this study could be more connected to each other. It's reasonable to assume that shade tolerance and mycorrhizal associations of host trees could affect the interactions between plant and soil pathogens, but how they influence the effects of insects and vertebrates on host performances? That is to say, the authors may need to provide more theoretical basis from the literature to expound that why different mycorrhizal types of host trees may influence the risk of insect and vertebrate attack.

Reviewer #3 (Remarks to the Author):

This study uses natural enemy removal experiments in an old-growth temperate forest in Northeast China to quantify the role of different natural enemies in maintaining diversity through conspecific negative density dependence. The authors examine the effects of vertebrate, insects, and pathogens and examine how plant traits, specifically shade tolerance and mycorrhizae type, relate to the strength of CNDD. The results of this study advance a mechanistic understanding of understanding variation in the strength of CNDD among species. The revised manuscript is well-written and the analyses and interpretations are sound, however the Bray-Curtis index should be analyzed using multivariate analyses.

Minor comments/suggestions below:

Pg 3 L73. Specify which adult densities (e.g. conspecific)

Pg 4 L93-94. This sentence is unclear.

Pg 14 L347. led instead of "let"

Pg 15 L367. The Bray-Curtis Index is a multivariate response and requires multivariate analyses, such as a PERMANOVA. See:

Anderson, M. J. 2001. A new method for non-parametric multivariate analysis of variance. *Austral Ecology* 26:32-46.

Anderson, M. J., R. N. Gorley, and K. R. Clarke. 2008. PERMANOVA+ for PRIMER: Guide to Software and Statistical Methods. PRIMER-E, Plymouth, UK.

CLARKE KR. 1993. Non-parametric multivariate analyses of changes in community structure. *Austral Ecol* 18:117–143.

Reviewers' comments:

Reviewer #2 (Remarks to the Author):

I have reviewed this paper for another Nature journal previously and I am happy to find that the authors have corrected most of the issues I raised previously. The authors also tried their best to address the comments from other referees.

This study tests the strength of conspecific negative density dependence (CNDD) in a temperate forest, identify which groups of natural enemies are more critical drivers of CNDD, and whether mycorrhizal association and shade-tolerance could predict enemy-mediated CNDD. They show in field manipulated experiments that fungicide, and less importantly insecticide, promoted seedling recruitment and survival, and these effects were significantly associated with conspecific density. The authors also show that EM species and shade-tolerant species were more sensitive to adult conspecifics. All these findings are consistent with density dependent accumulation of natural enemies.

While similar experiments have been conducted in tropical forests, which may there are many things I like about this manuscript. This study provides a good investigation on CNDD in a temperate forest, with detailed dataset from a 3-year manipulate experiment, which could be a nice complement to the existing literature that mainly focused on tropics and subtropics. By manipulating the access of different types of natural enemies, the authors actually open the “black box” of CNDD to reveal the relative importance of various drivers.

We thank the reviewer for his/her continued appreciation of our work, and thoughtfulness throughout the peer-review process.

There are still a few things that I like less. First, as Reviewer 1 pointed out, although the natural enemy effects on seedling performance are significantly correlated with host density (i.e. density dependence), the overall effects of host density in control treatments (i.e. under natural conditions) are neutral for both survival and recruitment. That is to say, it's not *negative* density dependence. This does not change the main conclusions of the study, but I would suggest the authors to reevaluate the usage of the term “conspecific negative density dependence”. Maybe just “conspecific density dependence” would be more appropriate in this case.

We agree with the reviewer's suggestion. We have replaced the term “conspecific negative density dependence” with “conspecific density dependence” accordingly in our revised manuscript, specifying when we mean “conspecific negative density dependence” by spelling it out.

Second, the authors found that ectomycorrhizal species are more sensible to negative plant-soil feedbacks near conspecific adults, which is conflicting with many other recent studies that found EM species were less negatively affected by pathogens related to AM species (e.g. Bennett et al. Science 355: 181-184, 2017). Although the authors discussed this and provided on possible explanation, this still sounds unconvincing to me. Mycorrhizal associations could be established on seedling roots very quickly, especially with high conspecific adult density.

We agree with the reviewer that the differences among EM and AM species do not match our predictions and are initially somewhat puzzling. In response to the reviewer's comment that our suggested explanation was speculative, we sampled a number of seedlings from the most abundant species to quantify mycorrhizal colonization. Our results showed that root colonization by EM fungi of the current-year seedlings was generally low. Specifically, the percentage of root colonization of *T. amurensis* and *Q. mongolica* was $18 \pm 9\%$ and $11.67 \pm 2.7\%$, respectively. The other two EM species, *Ab. nephrolepis* and *P. koraiensis*, had no evidence of colonization. By contrast, the colonization rate of mature trees is much higher. For example, a recent study found that mycorrhizal colonization of adult trees in our forests, *T. amurensis* and *P. koraiensis*, was $59.2 \pm 8.3\%$ and $62.3 \pm 8.3\%$, respectively (ref. 1). These differences suggest that mycorrhizal associations take some time to develop, at least in our forest, and year-old seedlings benefit less from mycorrhizal colonization. We have clarified this in the discussion (Lines 217-222) and included the mycorrhizal colonization results as the new Supplementary Figure 2.

Two things are worth bearing in mind when interpreting the results of our mycorrhizal colonization surveys. First, the seedlings were collected towards the end of the growing season. Many of the seedlings that died in our study probably died earlier on in their ontogeny, and were probably even less colonized than the ones we collected. Second, mycorrhizal colonization is likely to be especially low in the low-light environments typical of the forest understorey – thus while colonization might be high under different circumstances, it is perhaps unsurprising it was low in the seedling assemblages we sampled.

Third, the two hypotheses tested in this study could be more connected to each other. It's reasonable to assume that shade tolerance and mycorrhizal associations of host trees could affect the interactions between plant and soil pathogens, but how they influence the effects of insects and vertebrates on host performances? That is to say, the authors may need to provide more theoretical basis from the literature to expound that why different mycorrhizal types of host trees may influence the risk of insect and vertebrate attack.

We thank the reviewer for this suggestion. We have added a sentence to make our second hypothesis to be more specific, now it reads,

“Specifically, AM trees are more sensitive to fungi-mediated conspecific density-dependence than EM trees, and the conspecific density dependence mediated by both plant-associated fungi and insects would be greater for shade-intolerant than shade-tolerant species.”

Reviewer #3 (Remarks to the Author):

This study uses natural enemy removal experiments in an old-growth temperate forest in Northeast China to quantify the role of different natural enemies in maintaining diversity through conspecific negative density dependence. The authors examine the effects of vertebrate, insects, and pathogens and examine how plant traits, specifically shade tolerance and mycorrhizae type, relate to the strength of CNDD. The results of this study advance a mechanistic understanding of understanding

variation in the strength of CNDD among species. The revised manuscript is well-written and the analyses and interpretations are sound, however the Bray-Curtis index should be analyzed using multivariate analyses.

As the reviewer pointed out, the Bray-Curtis index summarizes differences in multivariate space. In most applications the goal is to determine whether treatments (or gradients) are associated with distinct clustering of the multivariate response variable along any axis or combination of axes – i.e. differences in any direction in multivariate space are considered interesting.

We are using the index in a more targeted way however, similar to its use in a previous study (i.e. ref. 2). Specifically, we are estimating the differences between seedling assemblages in each treatment and the surrounding adults – basically, we are measuring distances to a specific point in ordination space rather than separation in any direction in ordination space. Using this method also allows us to capture the experimental design better because we can account for the grouping structure by using mixed-effects models.

Perhaps it is worth illustrating the difference with an example. If seedling assemblages in control and fungicide treatments are very different, but equally as distinct from the adults, a PERMANOVA analysis would indicate significant differences, but that would not get at our question of whether fungi contribute to the trajectory of seedling composition towards that of the adults. Indeed, we would end up just doing a post-hoc comparison of distances between the centroids of adult composition and seedlings in each of the treatments, and then comparing those differences between biocide treatments. Our analyses take us directly to that point, and therefore, we believe, are more appropriate for our current analysis. Hopefully this makes our rationale clearer.

To make our rationale clearer to the reader, we have added the following sentence (Line 377-379),

“This metric quantifies the effects of biocide treatments on the compositional development of the seedling community along the trajectory towards the surrounding adults.”

Minor comments/suggestions below:

Pg 3 L73. Specify which adult densities (e.g. conspecific)

Added.

Pg 4 L93-94. This sentence is unclear.

Thanks for pointing out this. We have rewritten this sentence, it now reads,

“Dots and bars represent the mean and SE of the observed values, which were calculated by adding model residuals to the predicted values. We averaged the observed values within bins to facilitate visualization (with bin width increasing as the conspecific crowding index decreases).”

Pg 14 L347. led instead of "let"

Changed, thanks.

Pg 15 L367. The Bray-Curtis Index is a multivariate response and requires multivariate analyses, such as a PERMANOVA. See:

Anderson, M. J. 2001. A new method for non-parametric multivariate analysis of variance. *Austral Ecology* 26:32-46.

Anderson, M. J., R. N. Gorley, and K. R. Clarke. 2008. PERMANOVA+ for PRIMER: Guide to Software and Statistical Methods. PRIMER-E, Plymouth, UK.

CLARKE KR. 1993. Non-parametric multivariate analyses of changes in community structure. *Austral Ecol* 18:117–143.

As we explain above, PERMANOVA would not directly address our key question – do treatments affect the dissimilarity of seedling species composition to that of the surrounding conspecific adults, and would make it more difficult to account for the grouping structure of our data. Therefore, we prefer to continue using our more targeted analysis.

References uses:

1. Zhou, M. *et al.* Nitrogen deposition and decreased precipitation does not change total nitrogen uptake in a temperate forest. *Science of The Total Environment* **651**, 32–41 (2019).
2. Bagchi, R. *et al.* Pathogens and insect herbivores drive rainforest plant diversity and composition. *Nature* **506**, 85–88 (2014).